# Detection of central and obstructive sleep apneas in mice: A new surgical and recording protocol

**Gabriele Matteoli**, **Sara Alvente, Chiara Berteotti, Dario Coraci, Viviana Lo Martire, Martina Lops, Elena Miglioranza, Alessandro Silvani, Emilia Volino, Giovanna Zoccoli, Stefano Bastianini** *

Department of Biomedical and Neuromotor Sciences, Alma Mater Studiorum, University of Bologna, Bologna, Italy

* stefano.bastianini3@unibo.it

## Abstract

Sleep apnea is a common respiratory disorder in humans and consists of recurrent episodes of cessation of breathing or decrease in airflow during sleep. Sleep apnea can be classified as central or obstructive, based on its origin. Central sleep apnea results from an impaired transmission of the signal for inspiration from the brain to inspiratory muscles, while obstructive sleep apnea occurs in the presence of an obstruction of the upper airways during inspiration. This condition leads to repetitive episodes of reduced oxygen and elevated carbon dioxide levels in the bloodstream, which entail both direct and indirect adverse effects on vital organs, especially the brain and heart. Basic research on animal models has been instrumental in advancing the understanding of disease mechanisms and pathophysiology, and in expediting the development of targeted therapies in several medical fields. Among animal models, mice are the mammalian species of choice for functional genomics of integrative functions such as sleep. Mice have long been known to show sleep apneas, but the classification of sleep apneas as central or obstructive in mice is technically challenging due to the small size of these animals. Here we present a method aimed at identifying central and obstructive sleep apneas in mice. This method involves the surgical implantation of electrodes for recording the electroencephalogram and nuchal muscle electromyogram, which are the gold standard to study the wake-sleep cycle, and for recording the diaphragm electromyogram, which allows the detection of diaphragm contraction. The method also includes the simultaneous recording of the above-mentioned biological signals and breathing inside a whole-body plethysmograph and the data analysis allows to score wake-sleep states and to detect sleep apneas and categorize them into central and obstructive events.

## Introduction

Sleep-related breathing disorders are common in both children (0.7-33%) [1] and adults (14-67%) [2]. In particular, sleep apneas, which are temporary interruptions of airway flow during sleep, may lead to repeated episodes of significant hypoxia and hypercapnia during sleep [3].

**Data availability statement:** All relevant data are within the manuscript and its Supporting Information files.

**Funding:** This work was co-funded by the European Union (NextGenerationEU) under the National Recovery and Resilience Plan (PNRR) within the framework of the project "ESTROSA: Energy-autonomous System for TReatment of Obstructive Sleep Apnea" (Code MUR P2022RTCCA_001 – CUP J53D23014050001) and by the Italian Ministry of Health with the grant PRIN2022 for the project "CORSA: Chemogenetic and Optogenetic Rescue of Sleep Apnea in mice" (Code MUR 2022CR32TM_001 – CUP J53D2301094000) awarded to S.B. The funders had no role in study design, data collection and analysis, decision to publish, or preparation of the manuscript.

**Competing interests:** The authors have declared that no competing interests exist.

Sleep apnea can be classified as central (CSA) or obstructive sleep apnea (OSA), based on its origin. CSA arises from a breakdown in the brain signal for breathing, entailing the interruption of diaphragm muscle effort, while OSA occurs due to either complete or partial repetitive physical upper airway obstructions during inspiration despite continued or increased respiratory effort [3–5]. Sleep apnea may significantly impair cardiovascular, metabolic, and neurocognitive health [6–8].

Basic research on experimental animals has the potential to accelerate pathophysiological understanding and the assessment of drug safety and efficacy ([9–14], as recent examples) owing to the relative ease in performing mechanistic studies. Mice (*Mus musculus*) are the mammalian species of choice for functional genomics of integrative functions such as sleep. Mice are relatively cost-effective to acquire and maintain, have a small size, are easily manageable, and have a short lifespan and a rapid reproductive rate. Moreover, mice and humans show numerous similarities in the genomic organization and genetic traits (sharing approximately 99% of genes) [15], electroencephalogram (EEG) pattern and brain circuitry [16], and good conservation of neural mechanisms related to sleep and respiration [17,18]. This has led to the development and availability of a great number of different genetic or diet-induced mouse models of human disorders [19].

Notwithstanding these advantages offered by the study of mouse models, the study and classification of sleep apneas in such small animals encountered difficulties due to technical challenges and the longstanding belief that only CSAs are physiologically present in mice [20]. However, recent research has challenged this notion, shedding light on inspiratory flow limitation and hypoglossal nerve function in mice [21,22], and on the spontaneous occurrence of OSA-like events not only in a mouse model of Down syndrome (Ts65Dn mice) and of CDKL5 deficiency disorder (CDKL5 knock-out mice), but also in wild-type mice [23–25].

The growing focus on sleep apnea research in rodents is also reflected in the sharp increase in the number of impactful papers published over the past two decades. A significant portion of these studies has focused on the effects of intermittent hypoxia, a key characteristic of OSA, on various physiological systems [19]. In these experimental protocols, mice were exposed to repeated cycles of hypoxic and normoxic gas mixtures for periods ranging from a few days [26] to several weeks [27] to mimic the transient and cyclic episodes of arterial blood desaturation observed in OSA patients. Alternatively, other studies have employed invasive methods to create intermittent airway obstruction such as: inducing tongue enlargement in mice [28]; inserting inflatable balloons into the trachea [29]; applying nasal masks to repeatedly obstruct airflow [30]; or performing tracheostomies [31] in rats. Although these studies have offered invaluable findings on the pathophysiological consequences of OSA, they are inherently limited in studying the pathogenesis or possible therapies for OSA [19].

To explore the underlying causes of this respiratory disorder, various approaches have been employed to distinguish between CSA and OSA in genetically modified or diet-induced mouse models of sleep apnea [19]. For instance, Polotsky and co-workers introduced a non-invasive method to assess respiratory effort by equipping the WBP chamber with two air bladders, allowing for the detection of inspiratory flow limitations, a hallmark of airway obstruction [32].

Here, we provide a surgical, recording, and analytical protocol for distinguishing between CSA-like and OSA-like events in mice by simultaneously recording sleep states, breathing pattern, and diaphragm electromyographic activity inside a whole-body plethysmograph (WBP) [24]. Briefly, this method involves the surgical implantation of electrodes for the recording of the EEG and the electromyogram of neck muscles (nEMG) and the diaphragm activity (DIA), followed by the non-invasive recording of mouse breathing inside a WBP chamber for 7-8 hours. Finally, it also includes the discrimination of sleep stages based on EEG and nEMG

signals, and the classification of apneas into CSA and OSA based on WBP differential pressure and DIA signals.

We first confirmed the effectiveness of this method using adult C57BL/6J wild-type mice and, subsequently, we investigated a mouse model of Down Syndrome [24] and of CDKL5 deficiency disorder [25], which often involves sleep-disordered breathing in humans [33–35].

## Materials and methods

The protocol described in this peer-reviewed article is published on protocols.io, dx.doi. org/10.17504/protocols.io.yxmvme54og3p/v2 and is included for printing as supporting information file 1 with this article

This study was carried out in strict accordance with the recommendations of the European Directive 2010/63/EU for animal experiments and with Italian law. The protocol was approved by the Animal Welfare Committee of the University of Bologna, Italy (Legislative Decree n. 26 of 2014; authorization n. 779/2017-PR, n. 205/2019-PR, and n. 535/2022-PR). All surgery was performed under isoflurane anesthesia and all efforts were made to minimize the number of animals and their suffering in accordance with the ARRIVE guidelines.

### Surgery

As detailed in the S1 File and in the protocol published on *protocols.io*, the surgical implantation of the EEG, nEMG, and DIA electrodes was performed under aseptic conditions with the mouse anesthetized (isoflurane 2-2.5% in pure oxygen). Analgesic (0.1 mg of Carprofen) and antibiotic (3'750 I.U./mouse of benzathine benzylpenicillin and 1.5 mg/mouse of dihydrostreptomycin sulfate) therapy were administered subcutaneously at the beginning and the end of the surgery, respectively.

### Sacrifice

Animals included in the experiments described in this article were euthanized, in accordance with the authorized ethical protocol, either by decapitation under deep isoflurane anesthesia (4% in pure oxygen) or through transcardial perfusion with phosphate-buffered saline (PBS; pH 7.4) and 4% paraformaldehyde solution in PBS.

## Expected results

### Methods for categorizing sleep apneas

The first demonstration of the existence of upper airway obstruction in mice was established via an indirect approach in 2012 in unrestrained obese mice [32]. Hernandez and co-workers developed a non-invasive technique to assess the respiratory movements of the mouse, serving as a surrogate for the respiratory effort. Specifically, they equipped a WBP chamber with two separate air bladders, positioned in the animal chamber (sensor bladder) and beneath the chamber floor (reference bladder). During the recordings, the sensor bladder detects the movements of the mouse's chest and abdomen caused by respiration, while the reference bladder eliminates chamber pressure fluctuations resulting from the animal's tidal volume changes. Using this method, apneas can be classified as either CSA-like when the airflow cessation coincides with the lack of respiratory effort, or as OSA-like in the presence of diminished inspiratory flow, prolonged inspiratory time, and heightened respiratory motion [32].

This system has been extensively used by Polotsky and colleagues to quantify the respiratory effort in different mouse models of obesity induced by polygenic inheritance [32], leptin deficiency [36,37], diet [21,38], or serotonin deficiency [39]. Moreover, this non-invasive

approach has been instrumental in studying the involvement of the hypoglossal nucleus in OSA pathophysiology [22,40], and developing treatments for obesity hypoventilation [23,41–43] and for factors leading to its exacerbation (e.g., hypertension) [44]. Furthermore, it has been employed to explore the role of the nucleus of the solitary tract [45] and dorsomedial hypothalamus neurons [46] in regulating metabolism and breathing in obese mice.

However, despite the non-invasive nature of the method, the intensity of the recorded pressure signal is affected by the mouse's positioning on the air bladder, thereby preventing comparisons between periods when the animal changes its position. Therefore, to provide direct evidence of the diaphragm effort during OSA-like events, we developed a new method involving the surgical implantation of electrodes to record EEG, nEMG, and DIA activity inside a WBP chamber, allowing the simultaneous monitoring of the wake-sleep cycle, ventilation, and the electrical activity of the diaphragm, which is the principal inspiratory muscle. Using this approach, apneas can be qualitatively categorized as either CSA-like or OSA-like based on the combination of WBP differential pressure and DIA signals. As better explained in the "Materials and Methods" section, CSA-like events are identified by a simultaneous absence of activity in both the WBP and DIA signals, while OSA-like events are characterized by the absence of activity in the WBP signal accompanied by one or more bursts of activity in the DIA signal, indicating clear diaphragm activation.

In addition, the application of this method to adult C57BL/6J wild-type mice and Ts65Dn mice, a widely used mouse model of Down syndrome [47], revealed that the degree of airway obstruction during OSA can range from complete to partial, allowing for further differentiation between OSA and sub-obstructive (sub-OSA) events [24]. This sub-classification involves estimating the inspiratory airflow, calculated as the ratio between the tidal volume and the duration of the DIA activity. Following the latest version of the *American Academy of Sleep Medicine Manual for the Scoring of Sleep and Associated Events*, revised in February 2023 [48] confirming the previous definition of hypopnea [49], we identified sub-OSA events as breaths characterized by at least a 30% reduction in inspiratory airflow compared to the baseline value assessed on the previous breathing event.

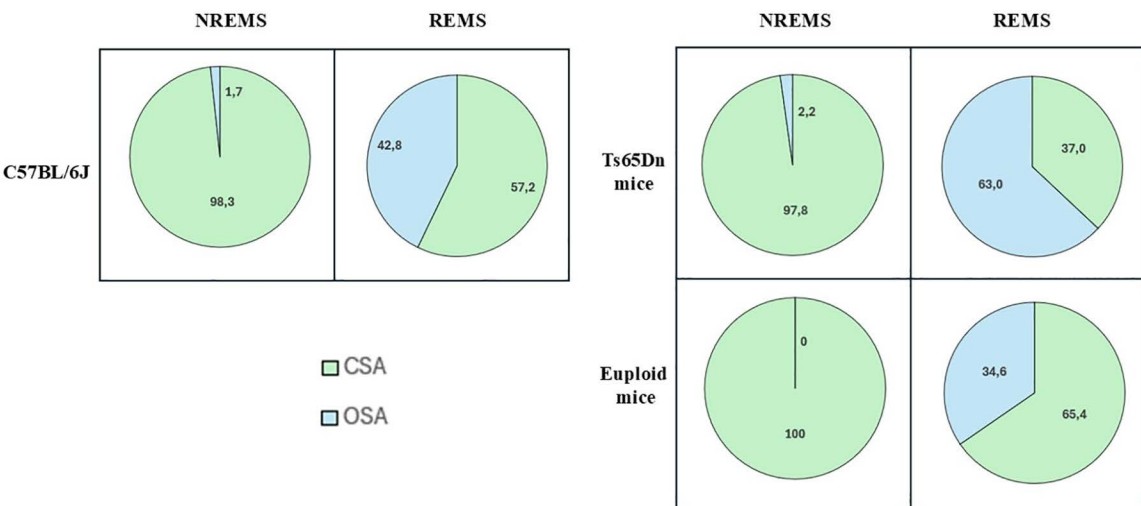

**Fig 1. Percentage of central and obstructive apneas during sleep.** Comparison of the percentage of sleep apnea of central (CSA) and obstructive (OSA) origin among C57BL/6J mice (n = 9), Ts65Dn mice (n = 9), model of Down syndrome, and euploid control mice (n = 11), recorded during non-rapid-eye-movement sleep (NREMS) and rapid-eye-movement sleep (REMS) inside the whole-body plethysmography chamber. Data are also presented in *Bartolucci et al. 2021* [24].

## Prevalence of CSA and OSA in wild-type mice and in a mouse model of Down syndrome

The following data are extracted from *Bartolucci et al., 2021* [24], where this surgical and analytical protocol was first validated in 9 adult C57BL/6J wild-type mice, and then applied to 9 Ts65Dn mice, a model of Down Syndrome, and 11 euploid controls.

Our data indicated that almost all apneas during non-rapid-eye-movement sleep (NREMS) were identified as CSAs, independently from the animals' genotype. Conversely, during rapid-eye-movement sleep (REMS), CSA and OSA were both present with a relatively higher contribution from OSA in Ts65Dn mice (Fig 1). Similar results were obtained in male mice lacking CDKL5, a model of CDKL5 deficiency disorder [25].

## Example of raw traces

Figs 2 and 3 show examples of raw traces of WBP differential pressure, DIA activity, EEG, and nEMG signals corresponding to CSA or OSA events, respectively, recorded in mice during NREMS or REMS.

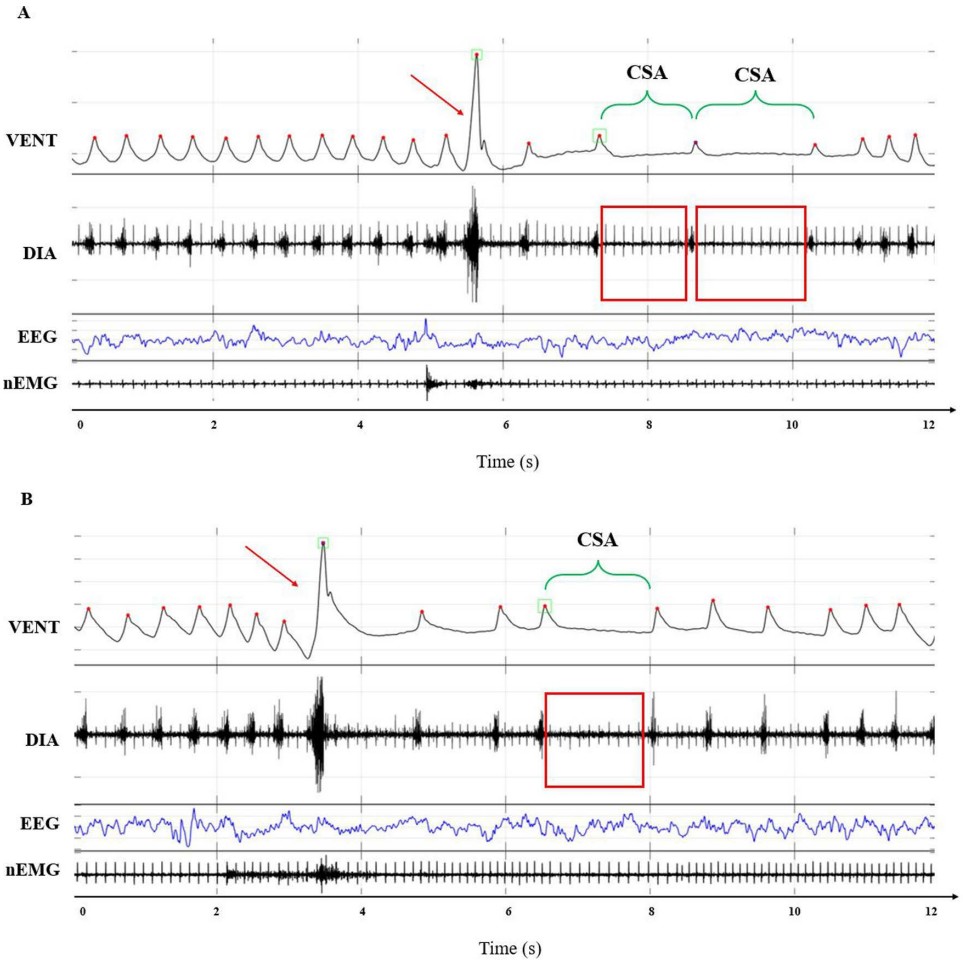

**Fig 2. Examples of central sleep apneas.** Panels A and B show representative examples of raw tracings corresponding to central sleep apneas (CSA). Each panel shows the differential pressure signal (VENT, corresponding to mouse respiratory patter), the diaphragm electromyogram (DIA), the electroencephalogram (EEG), and the nuchal electromyogram (nEMG).

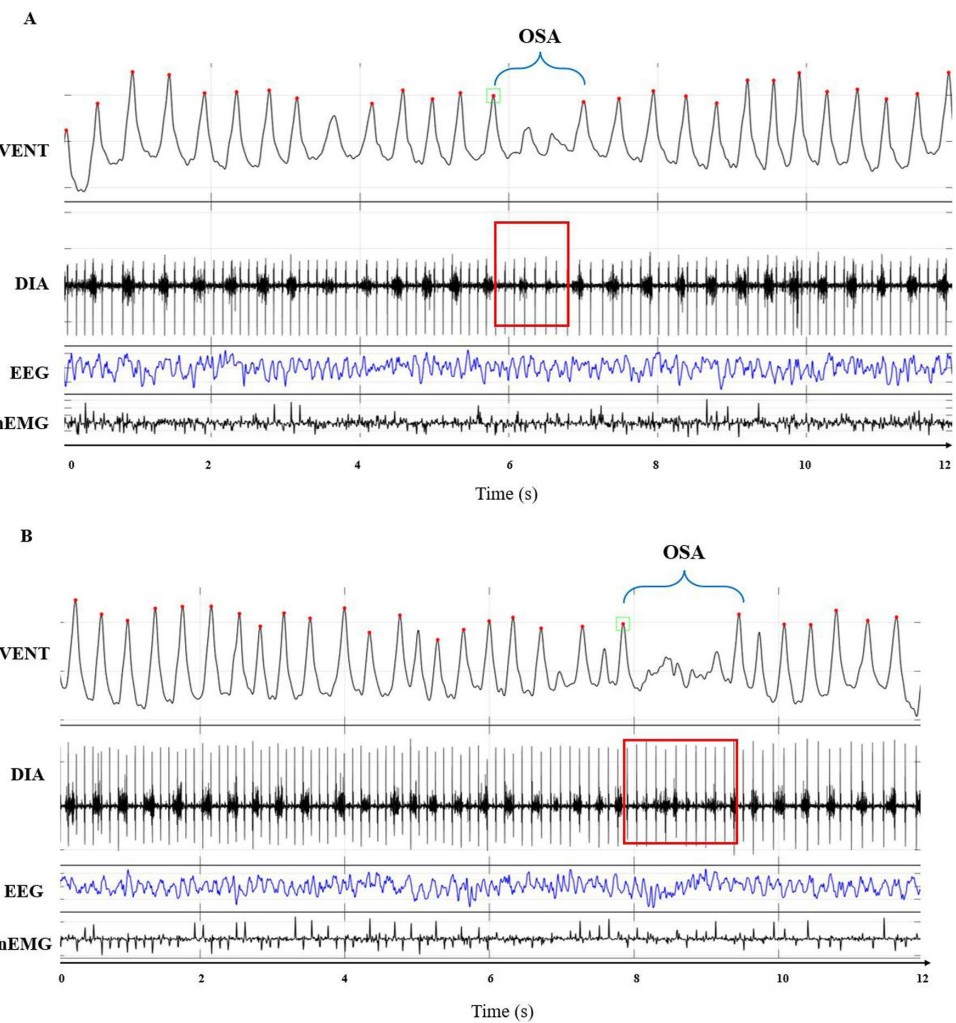

**Fig 3. Examples of obstructive sleep apneas.** Panels A and B show representative examples of raw tracings corresponding to obstructive sleep apneas (OSA). Each panel shows the differential pressure signal (VENT, corresponding to mouse respiratory pattern), the diaphragm electromyogram (DIA), the electroencephalogram (EEG), and the nuchal electromyogram (nEMG).

In the VENT signal tracing, upward deflections indicate inspiration, red dots indicate the peak of each inspiratory act, red arrows indicate sighs, green squares indicate breaths detected as apneas or sighs, and green brackets indicate the duration of the CSA. On the DIA signal, red rectangles indicate the absence of the DIA burst activity.

On the VENT signal, upward deflections indicate inspiration, red dots indicate the peak of each inspiratory act, green squares indicate the beginning of the apneic event, and blue brackets indicate the duration of the OSA. On the DIA signal, red rectangles indicate the DIA burst activity.

## Conclusions

This protocol provides a novel method for detecting CSA-like and OSA-like events in mice. Despite the invasive nature of the surgery and the requirement for basic surgical skills from researchers, the protocol ensures rapid (5-7 days) and complete post-operative recovery for the mice along with straightforward data analysis. Moreover, this technique offers flexibility,

as electrode crafting can be easily adapted for mice of varying ages and sizes. This protocol also holds the potential to be integrated with the existing intermittent hypoxia protocols, further advancing OSA research by addressing gaps or overcoming limitations identified in past literature. Finally, this protocol may serve as a valuable tool for gaining a deeper understanding of the disease mechanisms associated with sleep-disordered breathing in mice and might be functional in developing innovative therapies.

## Supporting information

**S1 File. Step-by-step protocol. Step-by-step protocol, also available on *protocols.io*.** (PDF)

## Author contributions

**Conceptualization:** Stefano Bastianini, Gabriele Matteoli, Sara Alvente, Alessandro Silvani, Giovanna Zoccoli.

**Funding acquisition:** Stefano Bastianini, Giovanna Zoccoli.

**Investigation:** Gabriele Matteoli.

**Methodology:** Stefano Bastianini, Gabriele Matteoli, Sara Alvente, Chiara Berteotti, Viviana Lo Martire, Elena Miglioranza, Alessandro Silvani, Giovanna Zoccoli.

**Project administration:** Stefano Bastianini, Giovanna Zoccoli.

**Resources:** Stefano Bastianini, Giovanna Zoccoli.

**Software:** Alessandro Silvani.

**Supervision:** Stefano Bastianini, Giovanna Zoccoli.

**Writing – original draft:** Gabriele Matteoli.

**Writing – review & editing:** Stefano Bastianini, Gabriele Matteoli, Sara Alvente, Chiara Berteotti, Dario Coraci, Viviana Lo Martire, Martina Lops, Elena Miglioranza, Alessandro Silvani, Emilia Volino, Giovanna Zoccoli.

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
