## [Decision Letter · Decision Letter 0]

22 Dec 2024

PONE-D-24-32191Detection of central and obstructive sleep apneas in mice: a new surgical and recording protocolPLOS ONE

Dear Dr. Bastianini,

Thank you for submitting your manuscript to PLOS ONE. After careful consideration, we feel that it has merit but does not fully meet PLOS ONE’s publication criteria as it currently stands. Therefore, we invite you to submit a revised version of the manuscript that addresses the points raised during the review process.

We look forward to receiving your revised manuscript.

Kind regards,

Pasquale Tondo, MD

Academic Editor

PLOS ONE

“This work was co-funded by the European Union (NextGenerationEU) under the National Recovery and Resilience Plan (PNRR) within the framework of the project “ESTROSA: Energy-autonomous System for TReatment of Obstructive Sleep Apnea” (Code MUR P2022RTCCA_001 – CUP J53D23014050001) and by the Italian Ministry of Health with the grant PRIN2022 for the project “CORSA: Chemogenetic and Optogenetic Rescue of Sleep Apnea in mice” (Code MUR 2022CR32TM_001 – CUP J53D2301094000) awarded to S.B.”

Reviewers' comments:

Reviewer's Responses to Questions

**Comments to the Author**

1. Does the manuscript report a protocol which is of utility to the research community and adds value to the published literature?

Reviewer #1: Yes

Reviewer #2: Yes

2. Has the protocol been described in sufficient detail?

To answer this question, please click the link to protocols.io in the Materials and Methods section of the manuscript (if a link has been provided) or consult the step-by-step protocol in the Supporting Information files.

The step-by-step protocol should contain sufficient detail for another researcher to be able to reproduce all experiments and analyses.

Reviewer #1: Yes

Reviewer #2: Yes

3. Does the protocol describe a validated method?

Reviewer #1: Yes

Reviewer #2: Yes

4. If the manuscript contains new data, have the authors made this data fully available?

Reviewer #1: Yes

Reviewer #2: Yes

**5. Is the article presented in an intelligible fashion and written in standard English?**

Reviewer #1: Yes

Reviewer #2: Yes

6. Review Comments to the Author

Reviewer #1: Dear Authors,

I read the manuscript entitled: Detection of central and obstructive sleep apneas in mice: a new surgical and recording protocol

In this manuscript authors described a method that can identify central and obstructive sleep apneas in mice. This method involves the surgical implantation of electrodes for recording the electroencephalogram and nuchal muscle electromyogram, which are the gold standard to study the wake-sleep cycle, and for recording the diaphragm electromyogram, which allows the detection of diaphragm contraction. This method also allows to score wake-sleep states and to detect sleep apneas and categorize them into central and obstructive events.

Animal models are crucial tools in biomedical and preclinical research, enabling reliable assessment of the safety and effectiveness of new therapies before they proceed to human trials. In the case of Obstructive Sleep Apnea (OSA), they are particularly valuable for advancing our understanding of its pathophysiology.

1- The introduction of the article is brief and concise. I believe, it is important to emphasize in the introduction the significance of this protocol by comparing it with what is already established in the literature.

2- There are several important studies exploring Obstructive Sleep Apnea (OSA) in mice, particularly focused on how intermittent hypoxia (IH), a hallmark of OSA, affects various physiological systems. This protocol could significantly advance research on Obstructive Sleep Apnea (OSA) by addressing gaps or limitations present in current literature.

Reviewer #2: Dear Authors,

I read your manuscript with great interest since, as authors pointed out, the article deals with a topic that is still little known. I found very attractive your protocol and I'm very sure that it will be appreciated by the readers.

Furthermore the manuscript is well written and the results are very interesting.

Wishing you the best in your work.

7. PLOS authors have the option to publish the peer review history of their article (what does this mean?). If published, this will include your full peer review and any attached files.

Reviewer #1: No

Reviewer #2: **Yes**

---

## [Author Response · Author response to Decision Letter 1]

16 Jan 2025

Rebuttal letter to the editor and reviewers

We sincerely appreciate the two Reviewers for acknowledging the significance of our study and providing valuable feedback. We have revised the manuscript following their suggestions to enhance the clarity and strength of our scientific message.

Below, we provide a detailed response to each of the points raised. All revisions are highlighted in red in the file titled "Revised Manuscript with Track Changes."

Reviewer #1:

Dear Authors,

I read the manuscript entitled: Detection of central and obstructive sleep apneas in mice: a new surgical and recording protocol. In this manuscript authors described a method that can identify central and obstructive sleep apneas in mice. This method involves the surgical implantation of electrodes for recording the electroencephalogram and nuchal muscle electromyogram, which are the gold standard to study the wake-sleep cycle, and for recording the diaphragm electromyogram, which allows the detection of diaphragm contraction. This method also allows to score wake-sleep states and to detect sleep apneas and categorize them into central and obstructive events.

Animal models are crucial tools in biomedical and preclinical research, enabling reliable assessment of the safety and effectiveness of new therapies before they proceed to human trials. In the case of Obstructive Sleep Apnea (OSA), they are particularly valuable for advancing our understanding of its pathophysiology.

1. The introduction of the article is brief and concise. I believe, it is important to emphasize in the introduction the significance of this protocol by comparing it with what is already established in the literature.

2. There are several important studies exploring Obstructive Sleep Apnea (OSA) in mice, particularly focused on how intermittent hypoxia (IH), a hallmark of OSA, affects various physiological systems. This protocol could significantly advance research on Obstructive Sleep Apnea (OSA) by addressing gaps or limitations present in current literature.

We have expanded the “Introduction” section to emphasize the significance of intermittent hypoxia protocols in investigating the pathophysiological effects of obstructive sleep apnea in mice (lines 82-93). We also provided a brief overview of current methods employed to distinguish between CSA and OSA in mice, with a specific focus on the technique developed by Polotsky and colleagues (lines 94-99) and the approach employed in our laboratory (lines 100-108). These methods are further discussed in the “Expected results” section of the present paper. Finally, the importance of our method, which can be seamlessly integrated with existing intermittent hypoxia protocols, is also highlighted in the “Conclusions” section (lines 233-236). To address these requests, we have included the following citations: Farrè et al., 2003; Almendros et al., 2011; Schoorlemmer et al., 2011; Lebek et al., 2020; Allaband et al., 2021; Hu et al., 2021.

Reviewer #2:

Dear Authors,

I read your manuscript with great interest since, as authors pointed out, the article deals with a topic that is -still little known. I found very attractive your protocol and I'm very sure that it will be appreciated by the readers. Furthermore the manuscript is well written and the results are very interesting. Wishing you the best in your work.

We sincerely thank the Reviewer for the kind compliments. We truly appreciate his/her positive feedback on our manuscript and are grateful that he/she found our protocol interesting and valuable.

Editor for Journal requirements:

We reviewed and modified the manuscript according to the “Manuscript body formatting guidelines” and the “Title, author, affiliation formatting guidelines”.

In addition to the information on anesthesia and the efforts to minimize the suffering of the experimental animals already included in the manuscript (lines 118-123), we have expanded the "Materials and Methods" section to provide further details on the anesthesia and analgesia protocols used during surgery (lines 125-131) as well as the methods of euthanasia (lines 133-137).

3. Thank you for stating the following financial disclosure: “This work was co-funded by the European Union (NextGenerationEU) under the National Recovery and Resilience Plan (PNRR) within the framework of the project “ESTROSA: Energy-autonomous System for TReatment of Obstructive Sleep Apnea” (Code MUR P2022RTCCA_001 – CUP J53D23014050001) and by the Italian Ministry of Health with the grant PRIN2022 for the project “CORSA: Chemogenetic and Optogenetic Rescue of Sleep Apnea in mice” (Code MUR 2022CR32TM_001 – CUP J53D2301094000) awarded to S.B.”

The funders were not involved in the study design, data collection and analysis, decision to publish, or manuscript preparation. This statement has been included in the “Financial Disclosure statement” section (lines 23-24) and in the revised cover letter.

Our ethics statement is included in the "Materials and Methods" section of the manuscript (lines 118-123).

We have reviewed our reference list and added the following citations to address the requests raised by the reviewers: Farrè et al., 2003; Almendros et al., 2011; Schoorlemmer et al., 2011; Lebek et al., 2020; Allaband et al., 2021; Hu et al., 2021.

---

## [Decision Letter · Decision Letter 1]

23 Feb 2025

Detection of central and obstructive sleep apneas in mice: a new surgical and recording protocol

PONE-D-24-32191R1

Dear Dr. Bastianini,

We’re pleased to inform you that your manuscript has been judged scientifically suitable for publication and will be formally accepted for publication once it meets all outstanding technical requirements.

Kind regards,

Pasquale Tondo, MD

Academic Editor

PLOS ONE

Additional Editor Comments (optional):

Reviewers' comments:

Reviewer's Responses to Questions

**Comments to the Author**

1. Does the manuscript report a protocol which is of utility to the research community and adds value to the published literature?

Reviewer #1: Yes

2. Has the protocol been described in sufficient detail?

To answer this question, please click the link to protocols.io in the Materials and Methods section of the manuscript (if a link has been provided) or consult the step-by-step protocol in the Supporting Information files.

The step-by-step protocol should contain sufficient detail for another researcher to be able to reproduce all experiments and analyses.

Reviewer #1: Yes

3. Does the protocol describe a validated method?

Reviewer #1: Yes

4. If the manuscript contains new data, have the authors made this data fully available?

Reviewer #1: Yes

**5. Is the article presented in an intelligible fashion and written in standard English?**

Reviewer #1: Yes

6. Review Comments to the Author

Reviewer #1: I have reviewed the revisions made to the manuscript based on the reviewers suggestions and i believe no further changes are needed. I wish you the best of luck!

7. PLOS authors have the option to publish the peer review history of their article (what does this mean?). If published, this will include your full peer review and any attached files.

Reviewer #1: No

---

## [Editor Report · Acceptance letter]

PONE-D-24-32191R1

PLOS ONE

Dear Dr. Bastianini,

I'm pleased to inform you that your manuscript has been deemed suitable for publication in PLOS ONE. Congratulations! Your manuscript is now being handed over to our production team.

Kind regards,

on behalf of

Dr. Pasquale Tondo

Academic Editor

PLOS ONE